Old maids have more appeal: effects of age and pheromone source on mate attraction in an orb-web spider

Cory Anna-Lena anna_lena.cory@yahoo.de
Schneider Jutta M.
Zoologisches Institut, Universität Hamburg , Hamburg , Germany
Hedrick Ann
Electronic publication date: 2016 Apr 18
Publication date: 2016
Volume: 4
Electronic Location ID: e1877
Received 2016 Feb 3; Accepted 2016 Mar 10
Copyright: ©2016 Cory and Schneider
Copyright year: 2016
Copyright holder: Cory and Schneider
License: This is an open access article distributed under the terms of the Creative Commons Attribution License, which permits unrestricted use, distribution, reproduction and adaptation in any medium and for any purpose provided that it is properly attributed. For attribution, the original author(s), title, publication source (PeerJ) and either DOI or URL of the article must be cited.
License URL: https://creativecommons.org/licenses/by/4.0/

Keywords: Female sex pheromone, Sexual signal, Male mate choice, Oviposition, Female condition

Funding: The authors received no funding for this work.

==============================
Background. In many insects and spider species, females attract males with volatile sex pheromones, but we know surprisingly little about the costs and benefits of female pheromone emission. Here, we test the hypothesis that mate attraction by females is dynamic and strategic in the sense that investment in mate attraction is matched to the needs of the female. We use the orb-web spider Argiope bruennichi in which females risk the production of unfertilised egg clutches if they do not receive a copulation within a certain time-frame.

Methods. We designed field experiments to compare mate attraction by recently matured (young) females with females close to oviposition (old). In addition, we experimentally separated the potential sources of pheromone transmission, namely the female body and the web silk.

Results. In accordance with the hypothesis of strategic pheromone production, the probability of mate attraction and the number of males attracted differed between age classes. While the bodies and webs of young females were hardly found by males, the majority of old females attracted up to two males within two hours. Old females not only increased pheromone emission from their bodies but also from their webs. Capture webs alone spun by old females were significantly more efficient in attracting males than webs of younger females.

Discussion. Our results suggest that females modulate their investment in signalling according to the risk of remaining unmated and that they thereby economize on the costs associated with pheromone production and emission.

Introduction

Mating partners have to meet in order to mate and reproduce. This prerequisite may be a trivial aspect in group-living animals, but it can constitute a significant factor within the overall costs of sexual reproduction in other taxa (Kokko & Wong, 2007; Magnhagen, 1991). Encounters with sexual partners can be facilitated by a wide range of mechanisms, e.g., meeting at certain locations (Hoglund & Alatalo, 1995; Van Ginneken & Maes, 2005) or producing long-range signals that attract potential mates (Gerhardt, 1994; Roelofs, 1995). The latter can occur via different communication channels (visual, acoustic or chemical), but which modality evolved will depend on the distance the signal has to travel in addition to other biotic and strategic selection pressures on signal design (Endler, 1993). Here, we are concerned with chemical communication that is considered to be an evolutionary ancient modality demanding the lowest cost in comparison to acoustic and visual signals (Bradbury & Vehrencamp, 1998). However, this assumption is largely untested.

Both males and females can produce sex pheromones to attract mates (Tillman et al., 1999; Wyatt, 2014), but particularly mate attraction by females is poorly understood. In many moths and spiders, females produce volatile or substrate bound pheromones, and males follow these traces (Gaskett, 2007; Greenfield, 1981; Wall & Perry, 1987). Selection on male signal perception has led to astounding sensory sensibility and accuracy, as exemplified by the receptor density and functionality on the antennae of the male moth, Bombyx mori (Kaissling & Priesner, 1970; Steinbrecht, 1970). The solid understanding of natural selection on males as signal recipients is opposed by a rudimentary comprehension of the female perspective. It is largely unstudied but nevertheless debated, whether pheromones emitted for mate attraction by females can be classified as sexual signals that actively transmit information to potential mates or as cues exploited by males (Andersson, 1994; Harari & Steinitz, 2013; Williams, 1992; Zuk & Kolluru, 1998). We use the orb-web spider Argiope bruennichi Scopoli, 1772 to test predictions from the hypothesis that pheromones are costly sexual signals.

Umbers, Symonds & Kokko (2015) recognized that the emission of cues that reveal one’s presence and location will not only reach the intended receiver, but likely attracts parasitoids and predators in addition to mates (e. g., Arakaki, Wakamura & Yasuda, 1996; Noldus, Potting & Barendregt, 1991; Zegelman et al., 1993). Besides increased mortality, the attraction of an adversely high number of mates also involves costs (Arnqvist & Nilsson, 2000). In spiders, the presence of males may be costly because of kleptoparasitism (Erez, Schneider & Lubin, 2005), web damage (Anava & Lubin, 1993; Harari, Ziv & Lubin, 2009; Watson, 1986), mate guarding (Calbacho-Rosa, Cordoba-Aguilar & Peretti, 2010; Fromhage & Schneider, 2005; Herberstein et al., 2005), and sexual harassment (Foellmer & Fairbairn, 2003; Lubin, 1986; Robinson & Robinson, 1973). Hence, even if the physiological costs of pheromone production would be low, extrinsic costs could be strong enough to select for a strategic use of pheromones.

In many spider species, including A. bruennichi, females are under pressure to attract males soon after maturation because egg maturation advances regardless of whether or not females have copulated. Hence, females that have not been found by a male in time will produce unfertilised egg clutches (Welke & Schneider, 2012). Therefore, if virgin females use pheromones strategically they should increase signalling efforts with increasing number of days passed since maturation (here called age).

Following this line of reasoning, apart from their location, female spiders can be expected to inform males about receptivity, mating status and perhaps even quality by regulating attraction signals. There is ample evidence that male spiders distinguish between virgin and mated females (Schulte, Uhl & Schneider, 2010; Stoltz, McNeil & Andrade, 2007; Tuni & Berger-Tal, 2012; Watson, 1986). Indeed, in many species females seem to lose receptivity after mating (Herberstein, Schneider & Elgar, 2002; Rabaneda-Bueno et al., 2008), but they may regain receptivity when sperm supplies require replenishing (Perampaladas, Stoltz & Andrade, 2008). Some studies have shown that virgin females produce sex pheromones that are absent in mated females (Chinta et al., 2010; Jerhot et al., 2010; Schulz & Toft, 1993). However, it is unknown whether the emission of sex pheromones by mated females discontinues because females are in control or because males avoid sperm competition by transferring pheromonostatic compounds (Arnqvist & Rowe, 2005; Thomas, 2011).

Note that in spiders, pheromones for long-range attraction are volatile but seem not to be released by a specific gland, rather they are found in the body and the silk (Gaskett, 2007; Schulz, 2013; Schulz, 2004). Many studies demonstrated that chemical compounds on female spider silk contain important information for males (Johnson et al., 2011; Stoltz, McNeil & Andrade, 2007; Sweger et al., 2010; Tuni & Berger-Tal, 2012). Gaskett (2007) speculated that silk-based pheromones are less costly while the emission of body pheromones might be more adjustable. Beyond speculations, no studies on web-building spiders compared if female bodies and web silk differ in the efficiency of mate attraction.

Here, we investigated age-dependency of male attraction by pheromones associated with the web or the female body. We conducted a field experiment with the European wasp spider A. bruennichi, in which the sex pheromone is known and found on the body and web of virgin females (Chinta et al., 2010). Using the synthesized pheromone, Chinta et al. (2010) also found that male attraction was dose-dependent. The presence of a response to pheromone concentration might imply that variation occurs in nature and that females vary in their pheromone emission. We predict that the longer females remain unmated, the more they should increase their investment in sex pheromone emission to attract males from larger distances and to outcompete other females in the proximity. If female signalling is costly for A. bruennichi females, we would expect that old females regularly attract more males than necessary but accept the costs because laying unfertilized egg clutches is even more costly. Note that only one mating event is needed to fertilize all eggs (Schneider, Fromhage & Uhl, 2005) and that due to efficient mate plugging by males, females are limited to use sperm of two males at most (Nessler, Uhl & Schneider, 2007). Field experiments are ideal for testing this prediction since they not only allow the qualitative assessment of wide-range attraction but also provide information about the intensity of mate attraction by counting the number of male arrivals. Females would benefit from regulating the emission within a short time frame because the mating season only lasts for 3–4 weeks (Zimmer, Welke & Schneider, 2012). Strategic signalling implies that with increasing age and pressure to attract males, unmated females should emit more pheromones.

Following this, we hypothesise that (1) mate attraction by virgin females increases with post-maturation age. To further understand how females enhance mate attraction, we tested females and their webs separately. Modulating pheromone content of webs requires rebuilding of webs and is, hence, less rapid than direct emission from the body. Thus, we hypothesise that (2) females adjust pheromone production on their web later than on the female body when the need of signalling is strikingly increased.

Material & Methods

Collection and maintenance

We collected 59 juvenile and sub-adult Argiope bruennichi females on a natural meadow near Nebenstedt, Lower Saxony, Germany (N53°097255, E11°13019) on June the 30th 2012. They were housed under laboratory conditions at ambient temperature and natural photoperiod and were sprayed with water five days a week. Depending on their body size, females were kept in 250 ml or 500 ml cups that had a hole stuffed with cotton wool on the top. Sub-adult females were transferred into Perspex frames (35 × 35 × 6 cm) where they build a normal orb-web. Small juveniles were fed with approximately 20 Drosophila hydei two times a week. Large juvenile, sub-adult and adult females received three Calliphora two times a week (for exception see below).

Female body measurements

Sub-adult females were inspected for moults on at least five days a week. Moults to maturity were recognized by inspection of the genitalia. The external genitalia of adult females have a spoon-shaped structure, the scape. In the sub-adult stage, the scape starts to develop but is covered with a thin-skinned layer. Mature females lose this layer, and the scape is exposed. At maturity, females were allocated in a pre-set order to the three age treatments. On the first day after maturation females were weighed on a calibrated scale (Mettler Toledo AB54-S) with an accuracy of 0.1 mg. Each female was also weighed on the day of the test (test weight).

Spider size was determined after the experiments. Females were brought back to the laboratory where they were kept and well supplied with food and water until natural death occurred. After death, the first pair of legs was removed and photographed under a stereomicroscope. We measured the length of the two segments tibia and patella with the Leica IM500 Imaging software (version 4.0; Leica Microsystems Imaging Solution Ltd., Cambridge, UK). We used this length as an approximation for the overall spider size. Legs and other sclerotized body parts do only change through moulting but not between moults. As an approximation of female condition, we used the residuals of the regression of body weight at maturation or at the test day and size of the females (Schulte-Hostedde et al., 2005). To achieve a normal distribution, we log-transformed the data for test weight.

Pheromone source

To disentangle the female body or the web as the source of pheromones, we used the webs of sub-adult females as “neutral webs” because they do not contain pheromones (Chinta et al., 2010). In the “female body” trials, virgin females were placed into these neutral webs. Most females showed little activity on the web and seemed to accept it. However, bad weather conditions led to more activity such that some females either stabilized the web with silk threads or destroyed the neutral web (see “Data analysis”). For the “web” trials, freshly constructed webs by virgins were used without the female. Generally, the web and its producer female were used on the same day (exceptions arise when webs are destroyed before they can be placed in the meadow). We ensured that the web was freshly built on the day of the trial by destroying webs on the evening before the trial. Females built their webs in the early morning, and we had introduced a Calliphora fly into the frame to stimulate the production of a capture web.

Female age classes

Oviposition can already occur 10–16 days after maturation. We used this time window to define three age classes of virgin females. Age is given as the number of days that had passed since maturation. “Young females” were 1–2 days adult, “middle-aged females” 3–7 days adult (5.8 ± 1.4), and “old females” had matured 8 or more days ago (10.0 ± 2.1). The webs were produced by females that were 1–2 days old in the groups of young females, by middle-aged females of 5.0 ± 1.3 days and by old females of the average age of 10.0 ± 2.8 days. We used a 2-day time window for the category of young females because binary choice experiments in the laboratory have shown that males do not distinguish between sub-adult females and females younger than three days (Schneider et al., 2016). A plausible explanation for this result is that females do not produce the pheromone until the 3rd day after maturation. Nevertheless, females readily copulate with males on the day of sexual maturity (Welke, Zimmer & Schneider, 2012). An age of at least eight days was used for the category “old females” to include females with high pressure to mate because oviposition is very close and the risk of producing unfertilised eggs is very high. Each female was used once for only a single age class.

A comparison of mate attraction by females of different ages requires testing a matched sample of females from all age classes every day. This is important since conditions on subsequent days are rarely the same: males disappear from the mating pool as they get cannibalised and male activity depends on weather conditions. However, females mature very synchronously which makes it difficult to find young virgins towards the end of the mating season (Zimmer, Welke & Schneider, 2012). To achieve a balanced availability of young and old females during the test period, we included a short period of food restriction during the final instars for half of the females which received three Calliphora flies once instead of twice a week for about two weeks. Spiders are adapted to endure periods of prey shortage since they regularly occur in nature. A short hunger period does not negatively affect spider survival or fecundity. However, it is sufficient to postpone moulting into maturity for a few days. We payed attention to feed all females well shortly before adulthood to standardize hunger level and distributed females from the two feeding regimes across all three age classes to avoid a systematic bias.

To account for the possibility that differences in juvenile feeding history might have an impact on the condition at maturation (Hector & Nakagawa, 2012) and influenced our age treatments, we tested whether the body condition on the first day of sexual maturity differed between females of different age classes. We found no significant differences (Table 1) but in the web trial, there was a trend that females allocated to the “middle-aged” group initially had a higher condition than the others, while young females had a lower condition. However, this would be only problematic if we found that middle-aged females attracted the most males.

Table 1 Female weight and female size information within the age classes and statistical comparisons (ANOVA) of female weight and female size between the age classes.

	Adult weight (mg)a	Size (mm)b	Conditionc	
	N	Mean ± SD	N	Mean ± SD	N	Mean ± SD	
Female body trials	
Young	12	92.2 ± 25.5	10	6.2 ± 0.6	10	−5.6 ± 5.9	
Middle-aged	12	94.0 ± 25.3	10	6.1 ± 0.6	10	4.5 ± 17.3	
Old	12	83.4 ± 25.3	10	6.0 ± 0.6	10	1.5 ± 7.1	
ANOVA	F2,33 = 0.68, p = 0.5148	F2,26 = 0.35, p = 0.7053	F2,26 = 2.10, p = 0.1425	
Web trials	
Young	14	96.0 ± 27.5	11	6.2 ± 0.6	11	−6.6 ± 6.3	
Middle-aged	14	98.2 ± 27.9	11	6.2 ± 0.6	11	4.4 ± 16.4	
Old	13	80.1 ± 23.2	10	5.9 ± 0.6	10	2.9 ± 11.4	
ANOVA	F2,38 = 1.88, p = 0.1664	F2,29 = 1.22, p = 0.3105	F2,29 = 2.88, p = 0.0725	
Notes.

a The adult weight was measured shortly after maturity.

b For the size, we used the tibia-patella length (segments of the leg) as a good approximation for the general size of spiders.

c Residuals of the regression of body weight shortly after maturation and size of the females.

Naturally, body condition increases with age (ANOVA on the weight of females across the age classes: F2,26 = 21.44, p < 0.0001; ANOVA on the weight of females that produced the webs: F2,29 = 21.32, p < 0.0001) such as body weight does. This increase is due to feeding and egg-maturation. Hence, age and condition are confounding factors and cannot be separated. In an attempt to dissociate the effects of female age and condition, we tested the effect of female condition on male attraction separately within each age class.

Test preparation (field)

Field experiments were conducted between 12th and 26th July 2012 on a natural meadow in Buxtehude, Lower Saxony, Germany (N53°453995, E9°674402). Our experimental period covered most of the mating season, although some males were still active when the trials ended. We observed male activity a few days before the experiments started and picked the experimental areas according to similar male activity.

The design was balanced in that we tested females of all age classes on each test day. Females were transported in containers (volume of 120 ml) to the field site and the webs for transported within the frames in specially designed transport boxes to minimise damage. On the test meadow, the Perspex frames with webs of the study females (“web” trial) or non-pheromone webs (“female body” trial) were fixed with tent pegs to the ground in an upright position. To ensure that the web was not destroyed by tall grasses, a small area around the frame was flattened. The distribution of the frames on the meadow was random although we ensured that the distance between the frames and local adult females was larger than 50 cm. For the “female body” trial, we carefully placed the females into the prepared neutral webs after the frames were securely fixed on the ground.

Test run

“Female body” trials and “web” trials were conducted apart from each other to avoid pheromone contamination. Either the two web categories had a distance of at least eight metres, or they were put out at different times of the day. The order and position was reversed every day so that each category was tested at each time of the day and in different parts of the meadow. The distance of eight metres was dictated by the size and structure of the test meadow.

We observed each web for two hours and counted the number of male visitors. Since not all males found their way into the web, we also counted males that apparently attempted to reach the females or their webs for at least 5 min. An attempt was only counted when the male wandered around grass lying in close vicinity to the female. To avoid pseudo-replication, we collected male visitors for the duration of the test. The males were released a few metres away from the test area after the test run was ended. There was a half-hour break between subsequent test runs so that the males could acclimatize in the field.

Data analysis

Although we were especially interested in the effects of female age and pheromone source, we also analysed the effects of size, test weight, and condition. However, female weight and condition are correlated and produced very similar results. We therefore excluded female weight from the analysis.

Most likely due to windy and humid weather conditions, we had a high proportion of zero counts. We divided the analysis into two levels and firstly used a binary response variable namely whether or not males approached. Secondly, we explored the treatment effects on the number of male visitors and reduced the sample to only include females that had male visitors.

The design was originally balanced with each 42 webs and females, but some webs and neutral webs were destroyed before the tests could start. Overall, we could test 41 webs and 39 females on neutral webs. However, particularly on rainy days some females escaped from the neutral webs mostly and observation time was less than 120 min. Since, we found that most first male visitors (92%) appeared within the first hour of the test run, we discarded only 3 cases that were shorter than 60 min and retained 36 tests that lasted for at least one hour for the first analysis of male appearance as a binary variable. However, for the test of treatment effects on the number of male visitors, we considered only females and their webs that were observed for the full two hours and in which males visited, at least, the web or the female (n = 14).

Statistical analyses were performed with the program R version 3.0.3. All variables were tested for normal distribution with the Shapiro–Wilk test. We used parametric tests for normally distributed data and nonparametric tests for analyses of non-normal and paired data. All tests were two-tailed. Since only a part of the “female” and “web” data was paired, we performed a generalised estimating equation (GEE) to explore whether females or webs were more likely to attract males. We created a binary model with pheromone source as testing variable and defined female ID as a random term (grouping variable). The correlation structure was specified as “independent.” We simplified the model by excluding the variable “pheromone source” and compared whether the original model significantly differed from the simplified model by using the Wald test. A significant difference between both models would be indicative for a significant effect of the pheromone source on the probability of being visited by a male.

Results

Only 23 of 77 females or webs were visited by males during the 2h-observation period, most of them (22 of 23) within the first 60 min. Presumably, male activity was low due to very humid and windy weather conditions. We included all trials into the analyses unless stated otherwise.

Females on neutral webs attracted visitors with a significantly higher probability (14 of 36 females) than webs alone (8 of 41 webs) (GEE: X2 = 6.44, N = 77, p = 0.0111; Fig. 1). A paired comparison of the number of male visitors attracted by a given female in a neutral web and by her own web was not significant (Wilcoxon signed rank test: T = 58.5, N = 14, p = 0.3724) suggesting that pheromone emission from the body directly or via the silk of the web were not necessarily linked. However, the sample size was low as we only considered 14 pairs in which males visited, at least, the web or the female.

Figure 1 Proportion of visited females (black bars) and visited female webs (white bars) depending on the female age.

The solid lines (female body) and dashed lines (web silk) show significant results between age classes. ∗, significant differences.

Female body

We used Fisher exact tests for pairwise comparisons of the proportion of females on neutral webs visited per age class. Although the probability of male visits increased with increasing age class (Fig. 1), we only found a significant difference between young and old females (N = 24, p = 0.0361). Female fixed size (t-test: t27 = − 0.67, p = 0.5079) did not influence the probability of a male visit while female condition had significant positive effects (t-test: t27 = − 3.49, p = 0.0017, Fig. 2). Note that any potential effect of female body condition was confounded by age (see methods). In an attempt to separate age and condition, we tested the effects of female condition within the three age classes and found no significant differences in condition of females that were or were not visited (young females: no value; middle-aged females: t-test: t7 = − 0.32, p = 0.7554, old females: logistic regression: X2 = 0.99, N = 10, p = 0.3187).

Figure 2 Condition of females separated into the occurrence of male visitation.

Females and their webs attracted more males, when they were in good condition. ∗, significant differences. The condition is shown as the log-transformed residuals from the regression of weight and the size of the females. The dotted line marks the zero line.

Excluding females that were never visited, females received 1–4 male visits (median = 2) within two hours. Due to the unbalanced sample sizes within the age classes (young females were hardly visited at all), we dropped the analysis with the category “age class”. Instead, we used “post-mature age” as a continuous variable, but we did not find a significant correlation between age and the number of male visitors (Spearman rank correlations: rs = 0.49, N = 13, p = 0.0909, Fig. 3). Interestingly, while visitation rates did not correlate with female size (Spearman rank correlations: rs = 0.82, N = 13, p = 0.789), the number of male visitors significantly increased with female condition (Spearman rank correlations: rs = 0.68, N = 13, p = 0.0103).

Figure 3 Number of male visitors depending on female post-mature age.

The lines show the functional graphs for the “female body” data (solid line), the “web silk” data (dashed line).

Web silk

To compare the attractivity of webs without females, we used, as above, Fisher exact tests for pairwise comparisons of the probability of being visited between female age classes. Since the 14 webs of middle-aged females and the 14 webs of young females hardly attracted males at all (only a single web was visited in each of the two groups), we compared each of both categories only with the category “old females.” The tests revealed that webs of old females (6 of 13 webs) attracted males significantly more often than webs of young females and webs of middle-aged females (for both pairwise comparisons we used the same statistic: N = 27, p = 0.0329, Fig. 1). While size (t-test: t30 = 0.81, p = 0.4224) of the females that had produced the webs did not significantly differ between visited and non-visited webs, the effect of condition was significant (t-test: t30 = − 2.81, p = 0.0086; Fig. 2) suggesting that females in better condition produced webs that were more attractive to males. However, remember that females in better condition were significantly older (Spearman rank correlations: rs = 0.74, N = 32, p < 0.0001) and 8 of 10 webs came from old females.

Most female webs were visited by one or two males (80%), although the maximum was five. Visitation rates cannot be compared between the age classes because 80% of the visited female webs were from old females, as already mentioned above. As with the female body, we conducted the analysis with the continuous variable “post-mature age.” The post-mature age correlated positively with the number of males that approached the web (Spearman rank correlations: rs = 0.69, N = 10, p = 0.0271). From a post-mature age of 14 days onwards, some females were visited by more than two males (Fig. 3). Neither female size (Spearman rank correlations: rs = 0.09, N = 10, p = 0.8149) nor female condition (Spearman rank correlations: rs = 0.61, N = 10, p = 0.0613) were significantly related to the number of male visitors.

Discussion

Our field assays provided support for the hypothesis of strategic pheromone emission. In accordance with our prediction, females close to oviposition (old and heavy) were most successful in mate attraction. Females used both their body and their web to transmit pheromones, although the latter became more relevant with increasing age of the female. As oviposition came close, the attractivity of the web for mate attraction increased. Female fixed size estimated from sclerotized body parts seemed to be irrelevant for male attraction while it was indeed the body condition that caused increased numbers of males to visit a female and her web. However, because of the correlative nature between female age and female condition, it is difficult to separate both factors.

Our results are in accordance with model predictions by Umbers and colleagues (2015) in that females increase their investment in signalling if they age without having encountered a mate yet. Young virgin A. bruennichi females, particularly at the beginning of the mating season and under high densities (Zimmer, Welke & Schneider, 2012), are likely found haphazardly by the protandrous roving males and females may save the costs of pheromone production. This is suggested by the high frequency of mate guarding of sub-adult females in the field (Uhl et al., 2015). As females age, egg maturation progresses and by mating too late, oviposition will take place, albeit laying unfertilised eggs. Females should be under selection to avoid this fate. Thus, it is adaptive to increase signalling effort if no males have appeared within a few days after maturation. Such females may find themselves in a low-density patch or too late in the season when most of the males are already mated and consumed (Zimmer, Welke & Schneider, 2012).

Extrinsic costs of signalling can result from the attraction of parasitoids or an detrimentally high number of mates. Male presence on the web is known to have costs for female spiders because prey capture is reduced, and predation risk is increased (Herberstein, Schneider & Elgar, 2002). In nature, A. bruennichi females mate with one or two males (Zimmer, Welke & Schneider, 2012) and a single copulation is sufficient to fertilise all eggs (Schneider, Fromhage & Uhl, 2005). During copulation males plug the paired genital openings of females to prevent sperm competition. The mating plugs are highly effective so that the majority of females can store and use sperm of two males at most (Nessler, Uhl & Schneider, 2007). In our study, old females attracted many more males than needed for fertilisation and more than of the maximal number of sires. Furthermore, webs were especially used for mate attraction when oviposition was very close. In fact, those webs attracted up to five males. The larger surface of the web may increase pheromone distribution. Hence, females may use pheromones strategically and remain hidden for males if they do not benefit from mate attraction. We did not test whether old females build larger webs compared to young females. Therefore, further studies should consider that females might increase pheromone distribution by extending the web surface or by adding more pheromones onto the web.

While extrinsic costs are most likely present, it is unclear whether pheromone production in these spiders has physiological costs as well. If signal production involves physiological costs, females in better condition may emit more and would honestly signal their quality. A purely strategic use does not require honesty because it would only imply increased signalling with increasing age and pressure to attract males. However, it is difficult to separate female condition from female age because females become fat as egg maturation proceeds. Accordingly, both variables reflect the time pressure on females to secure fertilisation by increasing investment into signalling. A more important influence of age is confirmed by binary choice tests in A. bruennichi, in which males showed no significant preference when presented with a choice between two young females of the same age that differed in condition and size (Schulte, Uhl & Schneider, 2010). Investment in mate attraction may only become important at a later stage, while young females rather use their energy for feeding and producing eggs. Due to the low sample sizes and the low activity of males in that year, we did not find effects of condition within the class of old females. More experiments are necessary to investigate whether females of different condition but of the same age attract more males.

Our findings are inconsistent with the result from Chinta and Schulz (S Chinta & S Schulz, unpublished data; reviewed in Schulz, 2013) that the production of the sex pheromone (trimethyl (2R,3S)-methylcitrate) of A. bruennichi increases until the fourth day after the final moult and then decreases. Since male attraction by this sex pheromone follows a concentration-dependent matter (Chinta et al., 2010), the middle-aged females of our study should have attracted more males than the old females. An explanation for these inconsistencies could be that females close to oviposition attract males with another semiochemical than trimethyl (2R, 3S)-methylcitrate. At least one other female specific compound is known (Chinta et al., 2010). Whether or not increased mate attraction of old A. bruennichi females is based on trimethyl (2R, 3S)-methylcitrate has to be resolved in further studies.

The use of a different semiochemical shortly before oviposition could be based on a signal or cue mechanism. Old females may actively enhance signalling with a further, perhaps more potent pheromone, if the risk of laying unfertilized eggs outweighs the costs of attracting too many males. Alternatively, females may passively emit oviposition cues or stress hormones, if the risk of laying unfertilized eggs increased. Note that it is also adaptive for males to find virgin females close to oviposition because the probablility might be lower, that females die before they deposited their first egg sac (Rittschof, 2011).

Research on the praying mantis Pseudomantis albofimbriata (Barry, 2010; Barry, Holwell & Herberstein, 2010) suggests that pheromone production could be directly linked to egg production and hence be considered a cue rather than a signal. In a simultaneous choice test, male mantises displayed a preference for females that carried more eggs, which correlates with body condition. Given a choice between two females in equally poor condition, the males still picked the one with more eggs in their ovaries (Barry, 2010; Barry, Holwell & Herberstein, 2010). This link between pheromone release and fecundity could be mediated by juvenile hormone titers (Barry, 2010). This explanation is unlikely to apply to spiders with strong first male sperm precedence such as in A. bruennichi (Austad, 1982; Jones & Parker, 2008; Nessler, Uhl & Schneider, 2007; Uhl et al., 2014). Most females will mate shortly after maturation and are no longer visited by males, most likely because mated females stop producing pheromones (Chinta et al., 2010). Copulation with mated females promises low fitness returns for males since the genital openings are blocked with genital parts of the predecessor (Nessler, Uhl & Schneider, 2007) and unlike in mantises where a male constitutes a significant nutritional benefit (Barry, Holwell & Herberstein, 2008; Birkhead, Lee & Young, 1988; Maxwell, 2000), males of Argiope are very small and provide only a small addition to the diet (Blamires, 2011; Fromhage, Uhl & Schneider, 2003). However, egg maturation is a continuous process and independent of mating. Hence, females that mate shortly after maturation and are no longer interesting for males, still develop their eggs at much the same speed as unmated females. To understand the mechanisms of dynamic pheromone emission, we need studies that examine the underlying physiological processes.

Pheromone emission in spiders is poorly understood. The glands responsible for sex pheromone secretion and the mechanism to attach them to the web silk are still unknown. Just a few studies investigated if female spiders use both their body and their web for mate attraction and no study compared the efficiency. For wide range-attraction, volatile pheromones rather than contact pheromones are used (Kasumovic & Andrade, 2004; Olive, 1982; Riechert & Singer, 1995; Searcy, Rypstra & Persons, 1999). Prouvost and colleagues (1999) found that in the house spider Tegenaria atrica, the body cuticle and the web differ in some chemical compounds. Possibly, the importance of female or web pheromones for mate attraction depends on the mobility of the spider species. In orb-web spiders, females are sessile and mainly use volatile pheromones for mate attraction while web silk might give males more specific information about the female upon contact. In wandering spiders, silk, especially from draglines, may have a higher relevance for mate attraction and mate search than the silk of orb-webs (discussed in Baruffaldi et al., 2010).

As far as we know, this is the first study that compared the efficiency of pheromones transmitted by web silk and body in the field. Beyond that, our study complements the picture of mechanisms of mate attraction in spiders. In most studies, age-dependent mate attraction was explained by female receptivity (Klein, Trillo & Albo, 2012; Papke, Riechert & Schulz, 2001; Roberts & Uetz, 2005). We can exclude this explanation because A. bruennichi females are always receptive (Schneider et al., 2006). We can show that beside moths (Umbers, Symonds & Kokko, 2015), female spiders adjust mate attraction to the pressure of achieving a copulation. Future research should combine female age and mating status to test when females stop signalling. A. bruennichi females are polyandrous and benefit from attracting males after the first copulation.

Conclusion

We conclude that female signalling is a dynamic process and depends on the time spent unfertilised. Female signalling seems to be strategic implying the presence of costs. We found that the signal strength variation was adaptive and likely adjusted to the females’ needs.

Supplemental Information

Data S1 Raw data

Click here for additional data file.

We thank Tomma Dirks, Angelika Taebel-Hellwig, Gabriele Uhl, and Stefanie Zimmer for their crucial support during the experimental planning, the collection of spiders, and the experimental conduction in the field. Furthermore, we would like to thank Jasmin Ruch and two anonymous reviewers for helpful comments on the manuscript.

Additional Information and Declarations

Competing Interests

Author Contributions

Data Availability

The authors declare there are no competing interests.

Anna-Lena Cory conceived and designed the experiments, performed the experiments, analyzed the data, wrote the paper, prepared figures and/or tables.

Jutta M. Schneider conceived and designed the experiments, contributed reagents/materials/analysis tools, wrote the paper, reviewed drafts of the paper.

The following information was supplied regarding data availability:

Data can be found in Data S1.

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
