# Peer review of "Old maids have more appeal: effects of age and pheromone source on mate attraction in an orb-web spider"

_PeerJ, doi:10.7717/peerj.1877_

## Round 0.1 · original submission · Minor Revisions

Please revise, paying c!ose attention to the comments of the reviewers.

Reviewer 1 ·

Basic reporting

No Comments on this, authors did a great job.

Experimental design

Generally, I think this a well designed study. My only concerns are as follows:
-limited sample size for some of the tests (N=14).
-I'm not sure about the independence of these two experiments. Were the neutral webs constructed by females used in the body study? If so, did you keep track of the IDs of the web constructor and whether this had any affects on the results. I'm not sure this is a large issue, but could be helpful in the presentation of the paper.

Validity of the findings

The results and findings are sound and I think very interesting. I have a few comments on the experiment and conclusions.
-How do females respond to being placed on a web that isn't theirs? While I don't think this affects the overall results, could this affect their investment into body pheromone, since there is no silk pheromone of theirs?
-Could you test for differences in the slopes of the lines in Fig. 3? This could be particularly enlightening as to the investment into these pheromones with age.
-As I read this, I went back and forth (and so did the author in the discussion) as to whether females control the emission of the pheromone (line 399). This control is essential for your argument in the paper that pheromones are signals, not cues. I tend to think of them as signals, and evidence suggested in your paper on the control of pheromones after mating (lines 372-375). The case needs to be made stronger in the discussion as to the controlled and intentional variation in emission of pheromones.

Comments for the author

I have very few general comments on the paper. It was very well written, and I found few minor flaws. Here are a few suggestions on the rest of the paper:
-line 55: define sexual signal and outline what criteria you must see to satisfy this definition in A. bruennichi
-line 94: need to add "pheromone associated with" the web or the female body for this sentence to make sense.
-line 112: no need for the comma after "female body"
-line 412: phrase "is no side effect" doesn't make much sense here. I'm not following your argument. Try rewording and making your final conclusion a different way.

Reviewer 2 ·

Basic reporting

The manuscript is logical and flows well. The tables and figures are clear and support the results.

Experimental design

The experimental design is sensible and the analyses are appropriate.

Validity of the findings

Conclusions are clear and follow logically from the analyses.

Comments for the author

Line 40: what is ‘the general set of communication channels’?

Line 101: Is it necessarily from a longer distance, or just more effective in attracting those males within a particular range? Particularly if mate attraction is dose-dependent.

Line 135: ‘well-supplied’ – does this refer to food?

Line 157 and 160-163: 1-2 days is very young – do you have observations that indicate females this young will mate?

Line 187: Add ‘males’ after ‘most’

Lines 216-220: Were the female body and web trials always conducted in the same parts of the meadow, or were they also alternated? If the same part, Is there a possibility that one part of the meadow had more males present?

Line 237: I’m not sure I understand how the design was balanced with an unequal number of webs and females?

Line 314: suggest avoiding the word ‘fat’ as the measure of weight does not discriminate between fat reserves and eggs, etc.

Line 330:Note it is also adaptive for males to find virgin females close to oviposition.

Line 338: How many mates are females limited to?

Line 340: Are the webs of older females larger than young females?

Lines 345-6: This part of the sentence is difficult to follow. Suggest re-wording.

Line 399: This introduces the question of whether females are actively investing less in pheromone production or not. In other words, is the ‘signal’ a byproduct - of condition, for example - or is it truly a pheromonal signal that has evolved to attract males? This is more of a signal/cue distinction than anything else, as the resulting behaviour of the males (attraction to a female) is the same, but the evolutionary processes and potential costs to the female are different as, if it is a true signal, then females may be able to maximise the cost/benefit ratio. Lines 350-352 and 361-362 suggest it is a signal, but as noted by the authors, requires further experiments.

---

## Round 0.2 · Minor Revisions

There are just a few more changes required, as described below.

Line 102. "over other females" does not make sense. Is there another way to say what you mean?
Line 103. If it is costly, why should they attract more than necessary?
Line 360. The word "rather" does not make sense here.
Line 364. Remove the comma.
Line 441. "Depends", not "depend".

Please make these changes and return the manuscript. Thank you.

---

## Round 0.3 · accepted · Accept

Thanks for the swift changes! I am happy to accept this paper.